# Pup Recruitment in a Eusocial Mammal—Which Factors Influence Early Pup Survival in Naked Mole-Rats?

**DOI:** 10.3390/ani13040630

**Published:** 2023-02-11

**Authors:** Michaela Wetzel, Alexandre Courtiol, Heribert Hofer, Susanne Holtze, Thomas B. Hildebrandt

**Affiliations:** 1Leibniz Institute for Zoo and Wildlife Research, Alfred-Kowalke-Str. 17, 10315 Berlin, Germany; 2Department of Veterinary Medicine, Freie Universität Berlin, Oertzenweg 19b, 14163 Berlin, Germany

**Keywords:** cooperative breeding, eusociality, offspring survival, naked mole-rat, *Heterocephalus glaber*, Bathyergidae

## Abstract

**Simple Summary:**

The naked mole-rat is a small, long-lived mammal that lives in large colonies in subterranean burrows. The lifestyle of naked mole-rats makes their study difficult, and many details about their biology remain enigmatic. For evolutionary biologists and ecologists, a special trait exhibited by this species, their eusocial lifestyle, is of particular interest. In this study, we explored the determinants of early offspring survival using data from observations of 14 captive colonies over a total period of seven years. Our study revealed that early pup survival was significantly improved by higher pup body mass and maternal number of mammae and significantly reduced by increased maternal body mass and colony size. The latter negative effect may, however, be an artifact of the captive conditions in which colonies were kept. We further discuss the implications of the similarities and differences of naked mole-rats to eusocial insects and cooperatively breeding mammals in these determinants to shed light on the origin and maintenance of eusociality in mammals.

**Abstract:**

In eusocial insects, offspring survival strongly depends on the quality and quantity of non-breeders. In contrast, the influence of social factors on offspring survival is more variable in cooperatively breeding mammals since maternal traits also play an important role. This difference between cooperative insects and mammals is generally attributed to the difference in the level of sociality. Examining offspring survival in eusocial mammals should, therefore, clarify to what extent social organization and taxonomic differences determine the relative contribution of non-breeders and maternal effects to offspring survival. Here, we present the first in-depth and long-term study on the influence of individual, maternal, social and environmental characteristics on early offspring survival in a eusocial breeding mammal, the naked mole-rat (*Heterocephalus glaber*). Similarly to other mammals, pup birth mass and maternal characteristics such as body mass and the number of mammae significantly affected early pup survival. In this eusocial species, the number of non-breeders had a significant influence on early pup survival, but this influence was negative—potentially an artifact of captivity. By contrasting our findings with known determinants of survival in eusocial insects we contribute to a better understanding of the origin and maintenance of eusociality in mammals.

## 1. Introduction

Many factors can influence offspring survival in mammals (Table 1), including individual [1,2], maternal [3,4], social [5,6,7] and environmental factors [1,2,3,4,5,6,7,8]. In particular, maternal characteristics are known to have a strong impact on early juvenile survival (Table 1) [1,3,9,10] because of the prolonged period of offspring nutritional dependence during gestation and lactation [9]. In mammals, offspring survival is, therefore, likely to depend on a breeding female’s ability to lactate and deliver adequate nutrients and energy through milk [11,12]. Whether this situation also applies to eusocial mammals is, however, an open question. On the one hand, studies from non-eusocial cooperatively breeding mammals show that in many cases, reproduction is monopolized by a dominant breeding pair [13], and brood care is provided by both parents and subordinate group members [14,15]. On the other hand, studies from eusocial insects show that in these species, parental investment is almost exclusively limited to brood production (egg-laying) by the queen(s), whereas all brood care is provided by helpers termed workers [16].

The African mole-rat family Bathyergidae is unique among mammals with respect to their variable social organizations, ranging from solitary to eusocial lifestyles [17,18]. The Damaraland mole-rat (*Fukomys damarensis*) and some of its social relatives in the genera *Cryptomys* and *Fukomys* were classified as eusocial by some researchers, while others disagree [16,18,19,20]. In contrast, the naked mole-rat (*Heterocephalus glaber*), with the highest reproductive skew of all social mole-rats, is the species with a lifestyle that most resembles that of eusocial insects and, therefore, unambiguously meets the original definition of eusociality [21,22,23]. Similar to eusocial diploid insects such as termites (Isoptera), naked mole-rat colonies exhibit cooperative brood care, a reproductive division of labor with a single, dominant breeding female (the queen), overlapping generations of several subordinate and reproductively suppressed colony members (non-breeders) [19] and distinct morphological and behavioral castes [24,25,26]. Naked mole-rats are, therefore, an appropriate species to study maternal as well as non-breeder effects on early pup survival and determine whether eusocial mammals have a distribution of parental care that resembles either that of other cooperatively breeding mammals or that of other truly eusocial species.

Although factors influencing offspring survival have already been investigated in most cooperatively breeding mammals [15], they have not yet been studied in detail in any eusocial mammal. Braude [27] showed that wild colonies of naked mole-rats have high rates of both recruitment and loss. Wild colonies increased by as many as 104 pups per year (mean ± standard error = 34 ± 25, *n* = 42 cases in which a colony was captured in successive years) and lost between 12% and 91% of their non-breeders per year (46 ± 23). As pups younger than two weeks do not leave their subterranean nest [28], early postnatal survival rates of wild pups remain unexplored. In captivity, pup survival is known to be highly variable and differs during various life stages [28,29]. As in other mammals [4,30,31,32], naked mole-rat pup survival is reported to be lowest during the first postnatal days [33,34]. Moreover, pup survival in naked mole-rats may be influenced by maternal and social factors [28,35,36]. For instance, Jarvis [28] showed that neonates may die within the first few postnatal days caused by maternal neglect and problems with milk production. Other colony members in addition to the queen may also influence survival of littermates. Non-breeders provide both direct and indirect pup care, e.g., huddling in the nest, grooming, donating caecotrophs, carrying food to the nest, maintaining and defending the burrow system [35]. They also were seen performing detrimental behaviors such as pushing, displacing or consuming pups [28,35,37]. Payne [37] observed that young were sometimes ignored or actually abused by non-breeders. Jarvis [28] further showed that a rapid increase in the number of non-breeders and frequent nest disturbances may also negatively affect early pup survival in captive colonies. It is, therefore, unclear what the net effect of non-breeders on pup survival is and how it compares with the effect of maternal care.

The aim of this study was to quantify early pup survival and identify factors influencing pup survival in the first 10 days postpartum in a eusocial breeding mammal, using data from seven years of observing captive naked mole-rat colonies. While the demographic and environmental conditions in captivity may not necessarily mimic those in the wild, the subterranean lifestyle would make such a study difficult, if not impossible, to replicate in the wild. In particular, the detailed monitoring of our captive colonies enabled us to disentangle the effects of (1) individual pup characteristics (sex, birth body mass), (2) maternal characteristics (body mass, number of mammae), (3) social characteristics (colony size, litter size) and (4) environmental characteristics (ambient temperature, ambient humidity, number of nestbox changes).

**Table 1 animals-13-00630-t001:** Effect of different variables on pup survival in mammals. The effect of variables on pup survival is denoted as positive (+), negative (−) and no significant effect (n.e.); studies were conducted under laboratory conditions (captive), natural conditions (wild) or fenced outdoor enclosures (semi-captive).

	Effect	Species	Condition	Sources
Pup birth body mass	+	*Fukomys mechowii*	captive	[1]
	+	*Ctenomys talarum*	captive	[2]
	+	*Peromyscus californicus*	captive	[38]
	+	*Oryctolagus cuniculus*	semi-captive	[6]
Pup age	+	*Leptonychotes weddellii*	wild	[39]
	+	*Fukomys mechowii*	captive	[1]
	+	*Meriones unguiculatus*	captive	[31]
	+	*Mus musculus L.*	wild	[40]
	+	*Oryctolagus cuniculus*	semi-captive	[6]
	+	*Heterocephalus glaber*	captive	[41,42]
Maternal body mass	+	*Clethrionomys glareolus*	captive	[4]
	+	*Oryctolagus cuniculus*	semi-captive	[6]
	n.e.	*Octodon degus*	captive	[43]
	n.e.	*Sigmodon hispidus*	wild/captive	[44]
	n.e.	*Peromyscus polionotus*	captive	[10]
Maternal age	+	*Leptonychotes weddellii*	wild	[39,45]
	−	*Heterocephalus glaber*	captive	[41]
Maternal behavior	+	*Peromyscus polionotus*	captive	[10]
	+	*Crocuta crocuta*	wild	[3]
Colony size	+	*Marmota marmota*	wild	[46]
	+	*Lycaon pictus*	wild	[5]
	+	*Suricata suricatta*	wild	[47]
	−	*Meriones unguiculatus*	captive	[7]
	−	*Arctocephalus gazella*	wild	[48]
	−	*Suricata suricatta*	wild	[47]
	n.e.	*Fukomys mechowii*	wild	[1]
	n.e.	*Meriones unguiculatus*	wild	[49]
	n.e.	*Canis lycaon*	wild	[50]
	n.e.	*Meles meles*	wild	[51]
	n.e.	*Myotis myotis*	wild	[52]
	n.e.	*Mirounga leonina*	wild	[53]
Litter size	+	*Rattus norvegicus*	captive	[54]
	−	*Meriones unguiculatus*	captive	[31]
	n.e.	*Fukomys mechowii*	wild	[1]
	n.e.	*Clethrionomys glareolus*	captive	[4]
	n.e.	*Octodon degus*	captive	[43]
	n.e.	*Mus musculus*	captive	[55]
	n.e.	*Peromyscus leucopus*	wild	[56]
	n.e.	*Marmota marmota*	wild	[46]
	n.e.	*Sus scrofa domesticus*	captive	[57]
	n.e.	*Canis lycaon*	wild	[50]
Ambient temperature	+	*Meriones unguiculatus*	captive	[49]
	n.e.	*Heterocephalus glaber*	captive	[58]
	n.e.	*Rhabdomys pumilio*	captive	[59]
Handling/disturbance	−	*Cryptomys hottentotus*	captive	[60]
	n.e.	*Heterocephalus glaber*	captive	[37]
	n.e.	*Rattus norvegicus*	captive	[61]
	n.e.	*Mus musculus*	captive	[55]

## 2. Materials and Methods

### 2.1. Study Species

Naked mole-rats are eusocial mammals that live in extensive, subterranean burrows with a mean colony size of 74 ± 49 individuals (*n* = 30 colonies) [62]. A colony usually contains one breeding female, the queen, who monopolizes mating by reproductively suppressing other colony members and is solely responsible for producing the colony offspring [19,63,64]. Naked mole-rats are long-lived, with a maximum recorded lifespan of 37 years [65]. Queens form long-lasting pair bonds with their breeding male(s) and are capable of sustaining reproduction for at least 27 years [66]. After a gestation of 72–77 days [35] the queen gives birth to 12 ± 5.7 pups (*n* = 84 litters, 1–29 pups) in the colony nest [28,41], in up to five litters per year [67]. Pups are born blind, with little locomotor activity and are kept warm in the communal nest by other mole-rats (Figure 1) [34,68]. Pups are exclusively nursed by the queen until weaning at ~40 days [28]. One week before they start to eat solid foods, pups begin to beg for caecotrophs from all colony members [34,37]. Naked mole-rats do not exhibit pronounced incest avoidance and may inbreed to a high degree (r = 0.63–0.88) [69,70] although they have a preference for mating with unfamiliar males [71]. Non-breeders usually are close relatives of the queen and her offspring and may, thus, increase their inclusive fitness by staying in the colony throughout their life and helping to rear the offspring [72].

### 2.2. Housing Conditions

The first naked mole-rat breeding stock (KI) used in this study was purchased from the ABQ BioPark Zoo (Albuquerque, NM, USA) in 2008. Based on this colony, nine more colonies (KII, W, I, H, L, J, K, M, G) were founded at our institute. In 2012, we purchased two other colonies—one colony (C) from the Osnabrück Zoo (Osnabrück, Germany) and one colony (N) from the Schönbrunn Zoo (Vienna, Austria). The breeding stock of colony (D) was composed of individuals from the colonies (KI and N), and colony (E) was founded by animals from the colonies (KI and C). This study presents data from 14 captive colonies closely observed for a total of seven years (September 2008–December 2015), each comprising a breeding pair as well as 29 ± 18 non-breeders. Observation periods of the colonies were as follows: 7–18 months (colonies: L, I, G, W, K, J, H, E, M), 34–39 months (colonies: D, C, N) and 76–87 months (colonies: KI, KII). Each colony was separately kept in an artificial burrow system inside a climatized box consisting of 5–8 acrylic glass boxes of 25 cm diameter and interconnecting tunnels. The ambient temperature was adjusted to 28.0 ± 1.25 °C via a heating coil around the burrow system and to a relative ambient humidity of 61.0 ± 13.8% using wet tissue paper inside the box lids, simulating environmental conditions similar to their native habitat [73,74]. The chambers were organized by the colonies themselves as nestbox, toilet(s) or food-store and contained wood shavings for bedding, unbleached paper tissue as nesting material and wooden branches and blocks for enrichment. Depending on the degree of soiling, boxes were cleaned and partly refreshed every day or every few days. Naked mole-rats were fed ad libitum with a mixed diet of vegetables and fruits, supplemented with high protein and vitamin enriched cereals. No free water was provided. All naked mole-rats were kept and bred at the Leibniz Institute for Zoo and Wildlife Research (IZW) in Berlin, with the approval by the local ethics committee of the ‘Landesamt für Gesundheit und Soziales’, Berlin, Germany (reference no. #ZH 156; 23 September 2008).

### 2.3. Data Collection

By inspecting all colonies for signs of delivery and newborn litters several times each day, we could record the birth of all litters within 12 h of postnatal life (Day 0). From the day of birth until an age of 80 days, all newborns were daily taken out of the nest, counted and weighed using a digital top-loading balance with a precision of 0.1 g (Kern 822–37, Gottl. Kern and Sohn, Albstadt, Germany). The colonies appeared relatively undisturbed by the regular checking procedure used to monitor the pups since no queen abandoned any of her neonates. Naked mole-rats remain, however, extremely sensitive to vibrations in and around their burrow system and, in our holding facility, responded to disturbances by hastily leaving the nest and ceasing their current behavior. If disturbances are too frequent, colonies usually move their communal nest to another place in the burrow. Thus, we used nestbox changing behavior within a colony as an indicator of stressful environment. To do so, we recorded every day (before the colonies were checked and cleaned) the location of the nestbox and toilet(s). We also recorded the ambient temperature and humidity in each colony system as proxies of the environmental conditions.

All newborns were marked within 12 h after birth, as previously described [28,75], and individual tissue samples were stored for later DNA extraction. Offspring sex was determined by a modified PCR protocol [76] following Szafranski et al. [77]. For permanent marking, all surviving pups received a transponder microchip at the age of three months (size 7 × 1 mm). The gestation period was 73.0 ± 3.6 days (*n* = 11 observed copulations) with a 6–11 day postpartum estrus (this study) [35]. If conception was not directly observed (*n* = 68 out of 79 litters) during behavioral estrus (2–24 h [28]), the conception date was defined as being 73 days before the birth of a litter. At conception, queens were weighed and their mammae numbers counted.

### 2.4. Statistical Analysis

The main goal of this study was to assess potential predictors of early pup survival in naked mole-rats (Table 2). The first days after birth appear to be the most critical days for pup survival (Figure 2B) [28,34,36]. We, therefore, summarized pup survival with a binary variable indicating whether neonates survived past their 10th day, including day 10, or not (i.e., scoring 1 when the pup survived from day 0 till 9, including day 9, and 0 otherwise). We used this variable as the response variable in a logistic regression model. We examined four categories of effects encompassing nine variables as predictors in the model. These predictors were organized as follows: (1) individual characteristics (pup sex and birth body mass), (2) maternal characteristics (body mass and number of mammae), (3) social characteristics (colony size, litter size) and (4) environmental characteristics (ambient temperature, ambient humidity, number of nestbox changes). All variables except pup sex were measured as quantitative variables. Other predictors were initially considered, including litter order, maternal body length, colony age, burrow length or number of related naked mole-rats to the queen, but they turned out to be highly collinear with maternal body mass and colony size and less revealing than the finally chosen predictors. The fitting procedure estimated a total of 10 parameters for the fixed effects (1 for the intercept and 9 for modeling the effect of the predictor variables). The analysis was performed at the level of the pup. After removing missing values from the entire dataset (16 queens and the 79 litters), 585 out of 869 pups of 15 queens from 57 litters were considered in these analyses. To account for these dependencies in the analysis, we considered both the identity of the mother and the identity of the litter as two random effects with a Gaussian distribution, which led to the estimation of two additional parameters, the two corresponding variances. We fitted the model using the function ‘fitme’ from the package ‘spaMM’ (version 4.1.2.) [78] in the program R (version 4.2.2.) [79]. To assess the statistical significance of each predictor, we carried out a likelihood ratio test comparing the goodness of fit between the full model and a model refitted after excluding the variable under consideration. The significance of the likelihood ratio tests was assessed by parametric bootstraps, which represent a more robust alternative to the traditional asymptotic likelihood ratio test. As estimates for fixed effects are expressed on the logit scale in logistic regressions, we converted the estimates into odds ratios in the text to facilitate interpretation. *P*-values are computed for 2-tailed tests. Correlations were calculated accordingly as Spearman’s rank correlations.

## 3. Results

### 3.1. Description of Early Pup Survival

During the seven years of study, 869 pups (79 litters) were born to 16 queens in 14 colonies (Figure 2A). The proportion of pups that lived more than 10 days was 57.0% (491 of 862 pups for which survival at this age was known). Survival did not change much after this point within the duration of the study, dropping to 50.6% (436 of 862 pups) by the age of 60 days (Figure 2B) and, therefore, corresponding to a survival rate of 88.8% (436/491) between day 11 and day 60, or an average 10-day survival rate of 97.6% (=((436/491)^(1/(60−11)))^10). Until day 10, all pups survived in 30.8% of the litters (24 of 78 litters for which survival was known for all pups) and all pups died in 14.1% of the litters (11 of 78 litters; Figure 2A). Seven queens successfully raised 100% of their litters and pups (*n* = 14 litters). The remaining nine queens had together at least one or more surviving pups in 83.1% of their litters (54 out of 65 litters). Four of these nine queens had 11 single litters in which all pups died (Figure 2A).

### 3.2. Predictors of Early Pup Survival

The statistical analysis of survival to the age of 10 days showed that individual, maternal and social characteristics all significantly influenced early pup survival (Table 2). Environmental characteristics had no significant effects (Table 2). The predictors with the largest effect sizes were pup birth body mass, maternal body mass, number of mammae and colony size (Table 2). Recomputing the regression slopes after standardizing these quantitative variables (with mean = 0, sd = 1) indicated that pup birth body mass showed the largest absolute standardized effect size (β = 2.06), followed by number of mammae (β = 1.56), colony size (β = −1.24) and maternal body mass (β = −1.00).

#### 3.2.1. Effect of Individual Characteristics

Individual pup characteristics had a highly significant effect on early pup survival (Table 2). Amongst these, pup birth body mass was the best predictor of early pup survival. Newborns (*n* = 843) weighed on average 1.97 ± 0.31 g (range 1.03 to 3.07 g). Heavier pups had a significantly higher chance to survive the first 10 days (Table 2, Figure 3C). Surviving pups had a birth body mass of at least 1.34 g. The model predicted that if birth body mass of pups would be increase by 0.5 g, the probability to survive would improve 27-fold. Pup sex did not significantly influence pup survival (Table 2).

#### 3.2.2. Effect of Maternal Characteristics

The survival of pups was significantly affected by characteristics of the mother (Table 2). The body masses of queens at conception varied widely from 27 to 65 g (50.0 ± 7.9 g). All breeding females showed fluctuating body masses at conception, with mass changes from −5 to +16 g (4.6 ± 4.2 g). Queens with a high body mass at conception were significantly less successful at raising pups than lighter queens (Table 2, Figure 3B). The model predicts that if maternal body mass would be decrease by 10 g, the probability for the pups to survive would improve 5-fold. Maternal body mass increased with the number of litters produced, a possible proxy for maternal experience (Spearman’s rank correlation, ρ = 0.375, *n* = 58, *p* < 0.001). Maternal body mass also increased with age (ρ = 0.486, *n* = 58, *p* < 0.001). All mammae were functional and their number remained constant for each queen. The number of mammae ranged from 11 to 13 (11.7 ± 0.9). For the 15 queens for which we knew the number of mammae, 13 had an odd number of mammae (86.70%) and 2 had an even number (13.3%). In 41.0% of the litters (*n* = 31 litters), the number of littermates exceeded or was equal (9.0%; *n* = 7 litters) to the number of mammae. Number of mammae positively influenced pup survival (Table 2, Figure 3A). The model predicts that with each additional mammae the probability for the pups to survive improve 9-fold.

#### 3.2.3. Effect of Social Characteristics

Pup survival significantly decreased with colony size (Table 2, Figure 3D). In our 14 established colonies, colony size ranged from 2 to 55 individuals. An increase in colony size and density resulted in decreased pup survival over the first 10 days. For each 10 additional mole-rats within the pre-defined size of the burrow system, our model predicts that pup survival would decrease 2-fold. The model further forecasts that pup survival dropped below 50.0% in colonies with more than 37 individuals (Figure 3D). Litter size ranged from 1 to 22 pups (11.0 ± 4.4), but we found no significant relationship between litter size and early pup survival (Table 2).

#### 3.2.4. Effect of Environmental Characteristics

Depending on the number of individuals in a colony, ambient temperature ranged from 24 °C to 30 °C (28 ± 1.25 °C). Ambient humidity varied between 10 and 90% (61.0 ± 13.8%) across colonies and during this study. The frequency of nestbox changes varied between colonies from 1 to 6 (2.1 ± 0.7) changes per week during the first 10 days postpartum. Despite such variation in these environmental variables, we were unable to detect any significant effect on early pup survival (Table 2).

## 4. Discussion

Our study of the determinants of early pup survival in naked mole-rats show that survival is variable and lowest in the first postnatal days (Figure 2B), similar to what was reported for small mammals [4,30,31,32], mole-rats [1,80,81] and other naked mole-rat colonies [28,34,36,82]. Body mass at birth showed the strongest effect on early pup survival (Figure 3C), with heavier pups exhibiting a higher chance to survive the first postnatal days than individuals with lower birth body mass. Comparable results (Table 1) were also reported for many other small mammals [1,2,6]. In naked mole-rats, all colony members tend to sleep and huddle together in the nest, with the breeding female typically assuming the nursing position at the top of the pile (Figure 1). Heavier naked mole-rat pups were observed to reach and remain on top of the pile more easily, therefore, close to the queen and her teats (Figure 1). This advantageous position may reduce their likelihood of being crushed or suffocated by other mole-rats in the nest.

Maternal characteristics also appear to be important predictors of early pup survival in naked mole-rats (Table 2). For eusociality to evolve in mammals, queens must allocate large amounts of energy and nutrients to offspring production during pregnancy and lactation. Hood et al. [12] showed that the milk of naked mole-rat queens is relatively dilute and low in energy compared to that of other rodents. Therefore, given the high reproductive output of breeding naked mole-rat queens (litter size: 1–29 pups) [83] and the absence of food provisioning by non-breeders before weaning, queens must produce an usually large amount of milk to meet nutritive requirements of their pups. Hood et al. [12] estimated, on the basis of pup masses and average litter sizes, a daily milk yield for lactating naked mole-rat queens of 30.5 g, equivalent to 58.0% of maternal mass. This is a large maternal investment, as a comparison with other rodents shows. For instance, a lactating mouse only produces an approximate daily milk yield of 14 g, equivalent to 35.0% of its body mass [84]. The view that maternal investment in milk production is critical in naked mole-rats is further supported by our finding that lactating queens with a higher number of mammae also had higher pup survival (Figure 3A). Among mammals in general, and rodents in particular, mean litter sizes usually are approximately one-half the number of mammae, and maximum litter sizes approximate mammary numbers [83,85]. Naked mole-rats appear to be an exception to this rule, as breeding females exhibit a maximum number of 14 mammae but can have litters of up to 29 pups [83]. In this study, in 50.7% (*n* = 38) of litters, the number of littermates was equal to or above the maternal number of mammae. So, as in other mammals but unlike eusocial insects, the presence of maternal care provided to pups in early life seems critical in naked mole-rats.

A high body mass in lactating queens may increase the amount of milk produced and, thus, could be advantageous for offspring survival, as seen in other mammalian species [4,6]. This idea is not supported by our results, which instead showed that queens with a higher body mass were less successful at rearing pups than light mothers (Figure 3B). The negative effect of maternal body mass may be real, but it could also be indirectly caused by the effect of other maternal characteristics.

A prime suspect for such a confounding variable is age. For age to qualify as a confounding variable, variation in age should be present between queens, larger queens should be older, and older queens should provide less maternal care to their pups. These three conditions seem to be satisfied in naked mole-rats. First, although we know the exact age of only some of our queens, estimating queen ages on the basis of morphological characteristics [41,86] revealed that their age ranged from 1 to 20 years. Second, we observed that the body mass of queens was positively correlated to the age estimate, with almost all queens experiencing a gain in body mass, averaging 12.0% over the course of the study. Naked mole-rat queens are expected to gain mass as they age because they generally undergo fecundity-enhancing vertebral elongation, another unique characteristic of their biology [24,25,26]. Our study contributes little information about the third condition—that maternal care may be lower in older queens. We did not discover obvious lactation problems in any of our queens, but it remains possible that adequate milk supply decreased with maternal age. Buffenstein [41] reported that elderly naked mole-rat queens tended to have very large litters and that pup survival was extremely low, possibly because of inadequate maternal milk production. In sum, it is likely that aging is at least partly responsible for the observed negative effect of maternal body mass on early pup survival. Future studies could clarify whether age is the sole factor explaining why heavier queens produce pups that are less likely to survive than lighter queens. We recommend future studies to also record aggressive behaviors since heavier breeding females in rodents are sometimes more aggressive [87], and queen aggressiveness may detrimentally affect pup survival in naked mole-rats [28].

The effect of alloparental behavior and the effect of the number of non-breeders on offspring survival are quite variable across cooperatively breeding species. Some studies show a positive link [5,46,88], others no correlation [1,47] or even a negative effect [7,89]. In naked mole-rats, the effect of the number of non-breeders (i.e., colony size) on pup survival had not been quantified before. Alloparental care is not essential for early pup survival in naked mole-rats, since a breeding pair alone can rear pups to weaning in both the lab (this study: pup survival rate when raised by pairs = 89.69%; [28]) and the field [62].

Nevertheless, since naked mole-rats are highly inbred [70], wild colonies contain a large number of individuals up to 295 [90] and non-breeders are reproductively suppressed, often philopatric throughout their life and while performing pup-care, it seems reasonable to expect the number of non-breeders to have a positive effect on pup survival as demonstrated for the closely related Damaraland mole-rat (*Fukomys damarensis*: Jarvis and Bennett, 1993 unpubl.).

In our study, however, an increase in the number of non-breeders significantly reduced early pup survival (Figure 3D). Our results may be influenced by the laboratory settings. As also reported by other researchers, captive naked mole-rat colonies often cease pup recruitment after an initial phase of high pup survival, and litter survivorship remains higher when some mole-rats are regularly removed from the colony [28,62]. Perhaps negative density dependence emerges because naked mole-rats tend to huddle all together in a single nest with the lactating queen on the top of the pile (Figure 1). Regardless of the actual colony size, all our colonies had the same nestbox size, which implies that the operational density at the nest necessarily increased with colony size. With an increasing number of mole-rats, pups were observed more frequently to be displaced to the bottom of the pile, and thus, away from the preferred nursing position, or crushed in the nest by other colony members. As studies of radio-tracking wild naked mole-rat demonstrated that colony nests vary considerably in shape and size and that mole-rats use several nest-sites at the same time [90], this phenomenon is likely to be an artifact of captivity.

A second reason for the higher pup mortality rate with an increasing number of non-breeders may be the fact that non-breeding mole-rats are also frequently observed to carry out negative pup-care behaviors [34,35,91]. These behaviors cause a substantial number of offspring deaths and often result in the failure of the whole litter (this study) [28]. In unsuccessful litters, pups are constantly dumped, kicked and dragged out of the nest until they die from physical trauma [28,35]. Colony members were also seen to kill and consume pups (this study) [28,34,91]. The motivation or purpose of these negative pup care behaviors by older colony members in a eusocial species are still unexplored but do not appear to be driven by the environmental stress factors that we recorded [91]. Whether these behaviors occur in the wild or only in captivity remains to be determined.

A third explanation for the negative effect of the number of non-breeders on pup survival could be that the positive effects of these individuals are lower in captivity than in the wild. In wild colonies, the breeding pair and their newborns may benefit from behaviors that occur little in captivity because they are irrelevant or have little or no effect in captivity. In particular, in the wild, non-breeders perform the high risk tasks of foraging, defending the colony and maintaining the tunnel system [27]—all behaviors that are of no relevance in an artificial burrow system.

A fourth explanation could be that the effect of non-breeders may vary during different offspring life stages. Clutton-Brock et al. [47], for example, showed that in meerkats (*Suricata suricatta*), pup mortality was not significantly related to group size between birth and emergence but was important for the time between emergence and five months of age. In our study, the survival of naked mole-rat pups was negatively correlated with colony size in the first 10 days postpartum, so it remains possible that positive effects of non-breeders may be received by pups at a later age.

A fifth and final possible explanation is that the lower rate of offspring survival at a higher density of naked mole-rats in the burrow system may also be of adaptive value. While small and recently founded colonies will likely need to acquire a large workforce as fast as possible to be able to defend and expand the burrow system to find sufficient food, a large and established colony in an environment with scarce resources may benefit from a high pup mortality that retains only the fittest pups and decreases competition.

## 5. Conclusions

Our findings highlight the diversity of factors influencing early pup survival in a eusocial mammal. Naked mole-rats combine the characteristics of cooperatively breeding mammals and eusocial insects. Therefore, as expected, both maternal and social predictors exhibited a significant effect on early pup survival. Surprisingly, maternal body mass and colony size had a significant negative effect on early pup survival. Our results thus reveal that studies performed in captivity may underestimate the benefits provided by non-breeders and overestimate the costs of increasing colony size unless variable nestbox sizes are available.

## Figures and Tables

**Figure 1 animals-13-00630-f001:**
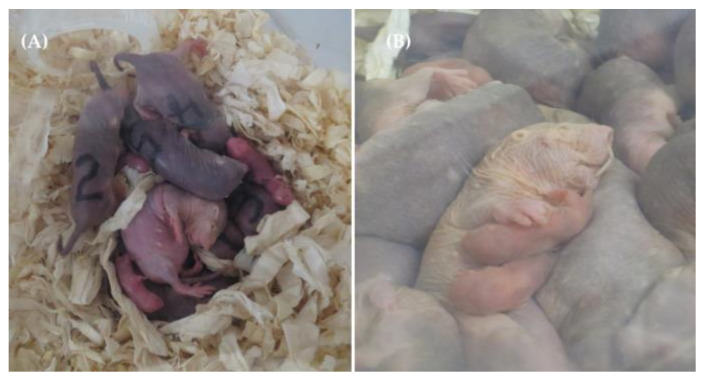
Naked mole-rat colonies huddling together in the nest, with the breeding female lying on the top of the pile. Picture (**A**) shows a small naked mole-rat colony (six non-breeders + queen + her pups) with enough space in the nestbox for the newborn pups to easily crawl around the huddle. Picture (**B**) shows a large colony (60 mole-rats) with the breeding female lactating her pups on the top of the crowded huddle.

**Figure 2 animals-13-00630-f002:**
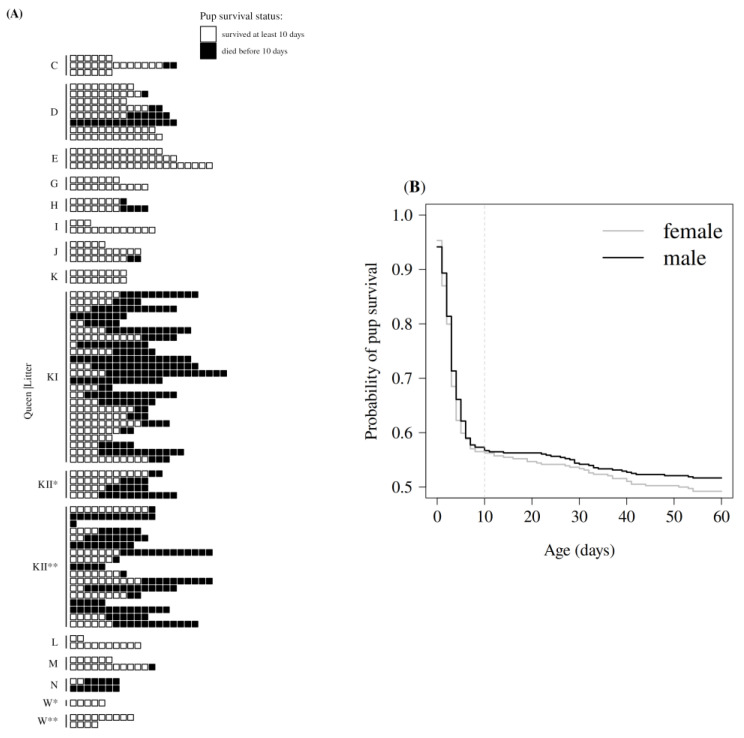
(**A**) Pup survival status at day 10 for 869 pups from 79 litters born to 16 queens in 14 colonies. Each row of squares represents one litter, with each square representing a pup. The color of the square refers to the survival status of the pup at day 10 as indicated in the legend. Litters grouped together correspond to the different litters produced by a given queen (e.g., C, D, E…). Asterisks label the first (*) and second queen (**) of a colony. (**B**) Survival curve of male and female naked mole-rat pups until 60 days postpartum. The curves represent the proportion of male (black curve) and female (grey curve) pups born that remained alive as time passed (i.e., Kaplan–Meier estimator). The day 10, used to determine the survival status of pups in the survival analyses, is indicated by a vertical, dashed line.

**Figure 3 animals-13-00630-f003:**
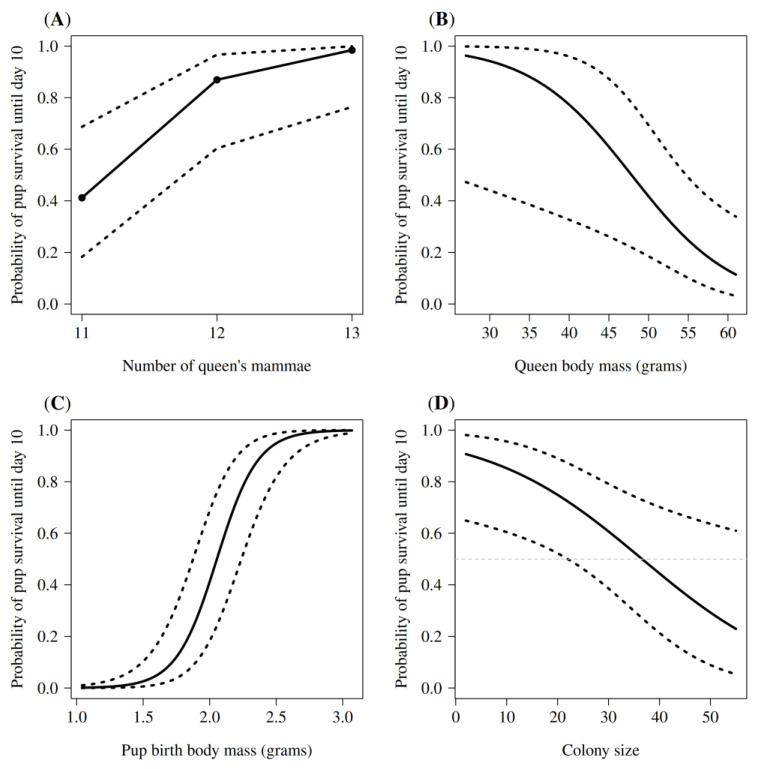
Predictions (black line) and 95.0% confidence intervals (black dashed lines) displaying the probability for naked mole-rat pups to have lived for longer than 10 days, i.e., to survive from birth (day 0) to day 9, as a function of: (**A**) maternal number of mammae, (**B**) body mass of queens at conception, (**C**) pup birth body mass and (**D**) colony size (i.e., number of all colony members) with the grey dashed line representing a 50.0% probability of survival on pup survival.

**Table 2 animals-13-00630-t002:** Summary table of the likelihood ratio tests used to compare offspring survival with respect to different predictors (*n* = 585 pups).

Predictor Variables	Beta	SE	χ^2^	df	*p*-Value
Individual characteristics			88.02	2	0.001
Pup birth body mass	6.55	0.90	88.01	1	0.001
Sex (male)	−0.22	0.33	0.53	1	0.453
Maternal characteristics			16.66	2	0.015
Queen body mass	−0.16	0.06	7.96	1	0.015
Queen mammae	2.25	0.87	6.31	1	0.048
Social characteristics			6.17	2	0.090
Colony size	−0.07	0.03	6.17	1	0.039
Litter size	0.09	0.09	0.91	1	0.423
Environmental characteristics			4.31	3	0.316
Temperature	0.26	0.58	0.25	1	0.653
Humidity	0.04	0.04	0.94	1	0.404
Nestbox change	−0.32	0.23	2.53	1	0.182

Note: Results presented as unstandardized parameter estimates (beta), the standard error for the unstandardized parameter estimates (SE), the chi-squared test statistic (χ^2^), the degrees of freedom (df) and the *p*-value (*p*).

## Data Availability

All data collected are available in the present paper.

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
