# Peer review of "Pup Recruitment in a Eusocial Mammal—Which Factors Influence Early Pup Survival in Naked Mole-Rats?"

_animals, 2023, doi:10.3390/ani13040630_

Round 1

Reviewer 1 Report

Line 57 – should be “The naked mole-rat (Heterocephalus glaber),.......”

Line 72 – should be “...34+25, n = 42......”

Line 97 – add the caveat that you recognize that demographics in captivity do not necessarily mimic those in the wild, but given that “its lifestyle makes it difficult to study it in the wild”, this is the best data available for understanding the problem at hand. Also, move lines 439-442 here. 

Table 1 – Put the footnote describing labels in the legend.  Single-space table and put it at the top of page 2.

Line 105 – delete scientific name (it is now above at first mention of the species)

Line 106 – change to “75+49”; nobody cares about 0.1 animals when the confidence limit is more than half the value

Line 115 – give a mean value for litter size

Line 121 – give an example of “high degree”

Line 135 – change to “29+18”

Line 138 – what is that symbol before “25 cm”?

Line 158-159 – “colonies appeared undisturbed” – add “To monitor this,....” and move part so lines 170-175 here.

Line 178 – “asses” should be “assess”

Line 179 – Table 2 should be placed before Figure 2 and nearer line 179 where it is first mentioned.

Line 181 – should be “..., including day 10, or not....”

Line 189 – tell us which variables each were correlated with so that readers know what to avoid

Line 214 – also report the survival rate during the interval from day 11 to day 60 (should be 50.6/57.0 = 88.8, or a 10-day rate of ~97.6)

Lines 295-296 – change to “Despite such variation in these environmental variables, we were unable to detect any significant effect on pup survival (Table 2).”

Lines 299-304 – delete or move to introduction

Line 447 – delete if no supporting information

Reviewer 2 Report

This paper analyzes the survival of pups in eusocial mammalian species, naked mole-rat. The object is unique, species was bred in different laboratories of the world for several decades, as it can serve as an object for analysis of many human health problems. The purpose of this study was to determine the causes of early pup mortality. In general, the results are close to those obtained earlier for different colonies and in the case of isolation of breeding females. Given the special status of this species, estimation of colony size and cage occupancy should be taken into account.

The main result was declared as " maternal body mass and colony size had a significant negative effect on early pup survival". In discussing the results, it was worthwhile to elaborate on some of the data. From the graph in Figure 2A, some litters were lost completely, some females (N, KI, KII*, KII**) lost more or almost as many pups as they saved. It would be interesting to trace the correlation of pup survival specifically for these females, for example, with breeding frequency (intervals between litters), physical condition, in particular, maternal body mass and general health characteristics, and especially female activity and skin condition. Usually, the quality of the fur is considered, but in naked mole-rat it is probably the skin that can serve as an indicator.

These remarks can be regarded as a recommendation.

Specific comments:

Simple Summary and Abstract present not exactly the same objective, accentuations, and conclusion:

line 18 "... to shed light on the origin and maintenance of eusociality in mammals." and lines 27-29 "... we present the first in-depth and long-term study on the influence of individual, maternal, social and environmental characteristics on early offspring survival in a eusocial breeding mammal"

The first section of "Material and Methods" (2.1) might be moved to the Introduction, as it gives a general overview of the species. In the next section (2.2. Housing conditions), in the description of the studied colonies, a reference to the origin of the colony founders is missing. However, there is data on genetic variability of this species, the range of which is rather large (Honeycutt, R.L., Nelson, K., Schlitter, D.A. and Sherman, P.W., 1991. Genetic variation within and among populations of the naked mole-rat: evidence from nuclear and mitochondrial genomes. The biology of the naked mole-rat, pp.195-208. Zemlemerova, E.D., Kostin, D.S., Lebedev, V.S., Martynov, A.A., Gromov, A.R., Alexandrov, D.Y. and Lavrenchenko, L.A., 2021. Genetic diversity of the naked mole‐rat (Heterocephalus glaber). Journal of Zoological Systematics and Evolutionary Research, 59(1), pp.323-340 and others). To compare results in future studies, it would be very helpful to specify the localities from which founders of colonies were derived, as in [79].

Figure 2. The legend is uninformative, both graphs should be described in more detail.

lines 311-312 - square brackets with species names are incorrect:

[Cryptomys mechowi: 1], Talas tuco-tuco [Ctenomys talarum: 2] and the European rabbit [Oryctolagus cuniculus: 6].

There are errors in the references, for example:

35 )

67. Brett, R.A., The Population Structure of Naked Mole-Rat Colonies, in The Biology of the Naked Mole-Rat, W.S. Paul, U.M.J. Jennifer, and D.A. Richard, Editors. 1991, Princeton University Press: Princeton. p. 97-136. DOI:10.1515/9781400887132-007. - should be Sherman, P.W., Jarvis, J.U. and Alexander, R.D., Editors

89 and 90 - duplicates - it is necessary to merge references
